# Reproducing topological properties with quasi-Majorana states

Adriaan Vuik[1*], Bas Nijholt[1] Anton R. Akhmerov[1] and Michael Wimmer[1,2]

**1** Kavli Institute of Nanoscience, Delft University of Technology,
P.O. Box 4056, 2600 GA Delft, The Netherlands
**2** QuTech, Delft University of Technology,
P.O. Box 4056, 2600 GA Delft, The Netherlands

★ adriaanvuik@gmail.com

## Abstract

Andreev bound states in hybrid superconductor-semiconductor devices can have near-zero energy in the topologically trivial regime as long as the confinement potential is sufficiently smooth. These quasi-Majorana states show zero-bias conductance features in a topologically trivial phase, mimicking spatially separated topological Majorana states. We show that in addition to the suppressed coupling between the quasi-Majorana states, also the coupling of these states across a tunnel barrier to the outside is exponentially different for increasing magnetic field. As a consequence, quasi-Majorana states mimic most of the proposed Majorana signatures: quantized zero-bias peaks, the $4\pi$ Josephson effect, and the tunneling spectrum in presence of a normal quantum dot. We identify a quantized conductance dip instead of a peak in the open regime as a distinguishing feature of true Majorana states in addition to having a bulk topological transition. Because braiding schemes rely only on the ability to couple to individual Majorana states, the exponential control over coupling strengths allows to also use quasi-Majorana states for braiding. Therefore, while the appearance of quasi-Majorana states complicates the observation of topological Majorana states, it opens an alternative route towards braiding of non-Abelian anyons and protected quantum computation.


# 1   Introduction

One-dimensional topological superconductors support Majorana bound states with zero energy at its endpoints [1–4]. Because of their non-Abelian exchange statistics and their topological protection to local sources of error, Majorana states are candidates for fault-tolerant qubits in quantum computing [5,6]. In addition to their non-Abelian properties, Majorana states have local signatures, namely $4\pi$-periodicity of the supercurrent in a topological Josephson junction [7,8], and a quantized zero-bias conductance peak in the tunneling spectroscopy of a single topological wire [9–11]. Because of the complexity of a braiding experiment demonstrating the non-Abelian statistics, experimental efforts so far focus on observing the local Majorana signatures [12–14].

An alternative explanation of the experimental observations is Andreev states with near-zero energy that appear in the topologically trivial phase [15–17]. These Andreev states can form at the wire's end, provided the confinement potential is sufficiently smooth [15]. Because smooth confinement potentials are likely to appear due to the separation between metallic gates and nanowires by dielectric layers, these quasi-Majorana states became a focus of recent theoretical research [18–23]. In particular, Ref. [18] shows that in case of smooth confinement potentials, trivial zero-bias conductance peaks are commonly appearing in Majorana devices, Ref. [21] demonstrates that near-zero energy Andreev bound states which are partially separated in space can reproduce quantized zero-bias conductance peaks, and Ref. [23] shows that such partially separated states can reproduce the fractional Josephson effect.

We demonstrate that quasi-Majorana states can be either partially separated or spatially fully overlapping, but in both cases these states have an approximately opposite spin. Because quasi-Majorana states are spin-polarised, the couplings across a smooth tunnel barrier within the WKB approximation are equal to:

$$\Gamma_{1,2} \propto \exp\left[-\frac{2}{\hbar}\int_{-w_{1,2}}^{w_{1,2}} |p_{1,2}(x)| dx\right],$$

$$p_{1,2}(x) = \sqrt{2m(E-(V_{\text{pot}}(x)\pm E_Z))},$$

$$\tag{1}$$

with $E$ the energy, $V_{\text{pot}}$ the potential energy, $E_Z$ the Zeeman energy, and $w_{1,2}$ the spin-dependent

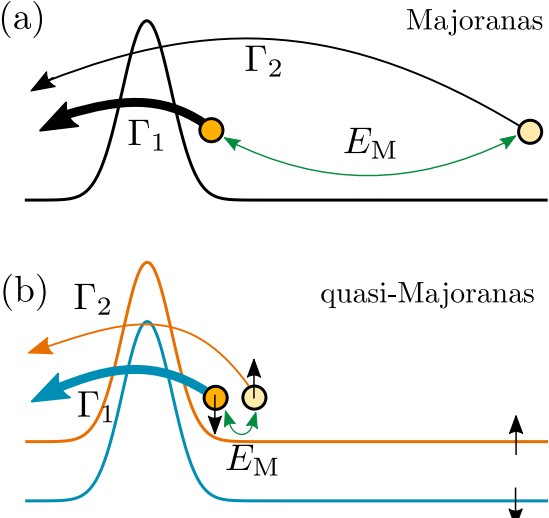

Figure 1: Schematic drawing of couplings of (quasi-)Majorana states. (a): Couplings in the topological regime with spinless Majorana states. The spatially separated Majorana states have a coupling energy $E_M$ (green arrow), and couple with coupling strengths $\Gamma_1, \Gamma_2$ across the tunnel barrier. The arrow thickness indicates that $\Gamma_1 \gg \Gamma_2$. (b): Couplings in the quasi-Majorana regime with two quasi-Majoranas, located at the tunnel barrier slope and with a suppressed coupling $E_M$, experience a different effective barrier due to their opposite spin and the finite magnetic field. The spin-down quasi-Majorana state (dark yellow) couples strongly with coupling strength $\Gamma_1$ (thick blue arrow, blue effective barrier), the quasi-Majorana state with spin-up (faint yellow) couples weakly with coupling strength $\Gamma_2$ (thin orange arrow, orange effective barrier).

width of the tunnel barrier. The ratio of the tunnel probabilities is

$$\Gamma_1/\Gamma_2 = \exp[-2(\gamma_1 - \gamma_2)], \tag{2}$$

where $\gamma_{1,2} = \int_{-w_{1,2}}^{w_{1,2}} |p_{1,2}(x)| dx$. Therefore, when the tunnel barrier is smooth and Zeeman splitting is sufficiently large, the quasi-Majorana couplings are exponentially different with the area of the barrier.

Such an exponential difference of the couplings, combined with the exponentially small coupling between the quasi-Majorana states, makes quasi-Majorana states indistinguishable from topological, spatially separated Majorana states, as we illustrate in Fig. 1. Because one of the two quasi-Majorana states is exponentially decoupled from the outside for increasing magnetic field in this regime, any local measurement will give the same result as for a truly topological system. We verify this phenomenon by analyzing tunneling spectroscopy of a quasi-Majorana device, the $4\pi$-periodic Josephson effect [24, 25], and a coupled quantum dot-nanowire system, which has recently been proposed [26, 27] and used [28] to measure Majorana non-locality. Because the exponential suppression of Majorana couplings requires a tunnel barrier, we then analyze the open regime and identify a quantized zero-bias conductance dip instead of a peak as a distinctive feature of topological Majoranas.

Because of the exponentially small coupling between quasi-Majoranas and of one quasi-Majorana across a barrier, a smooth tunnel barrier is an alternative approach to addressing individual Majorana states. As a consequence, braiding schemes can also be realized in a topologically trivial phase with quasi-Majoranas, since braiding effectively requires the coupling to a single (quasi-)Majorana state. Therefore, quasi-Majorana states supply an alternative route towards braiding non-Abelian anyons for quantum computing.

This paper is organized as follows. In Sec. 2, we describe our model and a method to compute coupling strengths. In Sec. 3, we discuss quasi-Majorana phase diagrams, wave functions, and couplings across a tunnel barrier. Sec. 4 describes quasi-Majorana effects on a coupled quantum dot-nanowire device and on a Josephson junction. We investigate an alternative local measurement in Sec. 5 and briefly discuss probing a bulk topological phase transition rather than local (quasi-)Majorana modes. To study quasi-Majorana states beyond a simple one-dimensional model, we show in Sec. 6 a phase diagram in a 3D nanowire with a smooth potential barrier. In Sec. 7, we discuss braiding with quasi-Majoranas. We give a summary and outlook in Sec. 8.

## 2 Model

### 2.1 Hamiltonian

We implement the minimum one-dimensional model, as proposed in Refs. [25, 29], with a Bogoliubov-De Gennes Hamiltonian given by

$$H = \left( \frac{p_x^2}{2m^*} - \mu + V_{\text{pot}}(x) \right) \tau_z - \frac{\alpha}{\hbar} p_x \sigma_y \tau_z + \Delta(x) \tau_x + E_Z \sigma_x, \tag{3}$$

with $m^*$ the effective mass, $p_x = -i\hbar\partial_x$ the momentum, $\mu$ the chemical potential, $V_{\text{pot}}$ the potential, $\alpha$ the spin-orbit interaction (SOI) strength, $\Delta$ the superconducting gap and $E_Z$ the Zeeman energy due to a parallel magnetic field. The Pauli matrices $\sigma_i$ and $\tau_i$ ($i = x, y, z$) act in spin and particle-hole space, respectively. The potential $V_{\text{pot}}$ and the position dependence of the superconducting gap $\Delta(x)$ vary for different devices, as specified in the following subsection. We choose the following parameter values of the Hamiltonian (3): $m^* = 0.015m_e$, corresponding to an InSb nanowire, $\alpha = 50$ meV nm, and $\Delta = 0.5$ meV, unless specified otherwise.

### 2.2 Devices

We implement the Hamiltonian (3) in three different devices, schematically shown in Fig. 2, that are used to measure local Majorana signatures. The system of Fig. 2(a) is a tunnel spectroscopy setup consisting of a proximitized nanowire of length $L_{\text{SC}}$ with a chemical potential $\mu$ and constant superconducting gap $\Delta$ connected on the left to a semi-infinite normal lead via a potential barrier $V_{\text{pot}}(x)$. The potential in this device is given by a Gaussian-shaped barrier, $V_{\text{pot}} = V_{\text{barrier}}$, with

$$V_{\text{barrier}}(x) = V e^{-(x-x_0)^2/2\sigma^2}, \tag{4}$$

with $V$ the height, $x_0 = 0$ the center and $\sigma$ the smoothness of the potential barrier.

Figure 2(b) shows the second system, a coupled quantum dot-nanowire device, which has been proposed recently as an additional tool for measuring the non-locality of Majorana states [26, 27]. Compared to the setup of Fig. 2(a), we replace the lead by a normal quantum dot ($\Delta = 0$) of length $L_{\text{dot}}$ with hard-wall boundary conditions at $x = 0$. The effective potential is $V_{\text{pot}} = V_{\text{barrier}} + V_{\text{dot}}$, with $V_{\text{barrier}}$ as given in Eq. (4) and $V_{\text{dot}}$ describing the chemical potential difference between the dot and the nanowire:

$$V_{\text{dot}}(x) = \frac{1}{2}\mu_{\text{dot}} \left( \tanh\left( \frac{x-x_0}{dx} \right) - 1 \right), \tag{5}$$

with $\mu_{\text{dot}}$ the chemical potential in the quantum dot, $x_0 = L_{\text{dot}}$ the interface between dot and nanowire, and $dx$ the length scale over which the chemical potential varies. This potential

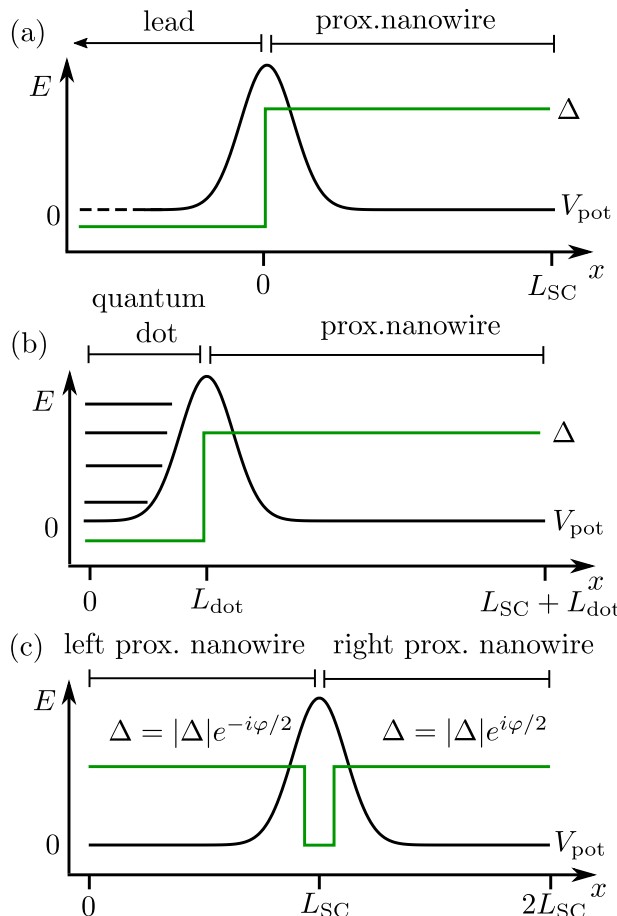

Figure 2: Schematic drawings of the three studied devices. The black lines indicate the potential profile $V_{pot}(x)$, the green lines the superconducting gap $\Delta(x)$. (a): Proximitized nanowire of length $L_{SC}$ with constant superconducting gap $\Delta$ connected to a semi-infinite normal lead from the left via a potential barrier. (b): Proximitized nanowire of length $L_{SC}$ connected to a normal quantum dot of length $L_{dot}$ on the left via a potential barrier $V_{pot}$. (c): Two finite proximitized nanowires, both of length $L_{SC}$, with a superconducting phase difference $\varphi$ between them, and separated by a potential barrier $V_{pot}$.

allows for different local chemical potentials in the dot and the wire, for example due to local gates in an experiment.

Finally, we consider a Josephson junction, consisting of two one-dimensional proximitized nanowires separated by a potential barrier and with a phase difference $\varphi$, see Fig. 2(c). In this device, $V_{pot} = V_{barrier}$, with the center of the potential barrier between both superconductors ($x_0 = L_{SC}$), and the position-dependent superconducting gap described by

$$\Delta(x) = \begin{cases} |\Delta|e^{-i\varphi/2} & x < L_{SC} \\ |\Delta|e^{i\varphi/2} & x > L_{SC}, \end{cases} \tag{6}$$

with a phase difference $\varphi$ across the junction. In all devices, we fix the nanowire length to $L_{SC} = 3 \ \mu$m. In the coupled quantum dot - nanowire device, we take a quantum dot length of $L_{dot} = 250$ nm.

In this work, we focus on quasi-Majorana states formed at the monotonously changing slope of a smooth tunnel barrier, specifically as given in Eq. (4). In particular, it should be noted that the quantum dot of the setup in Fig. 2(b) does not play a role in the appearance of

quasi-Majorana states – the smooth tunnel barrier slope on the side of the proximitized wire is essential. The tunneling to the dot rather serves as a probe of the quasi-Majorana state.

Though we are focusing on the specific case of a Gaussian potential barrier, previous work has found near-zero energy Andreev bound states also for different types of potentials: a linear potential [15], a quantum dot in the proximitized wire [18], and smooth potentials with some sharp features, such as point-like impurities or abrupt changes in the superconducting order parameter [15]. We expect our findings to be similar for such potentials, too.

We discretize the Hamiltonian (3) on a regular one-dimensional grid, and diagonalize this Hamiltonian to obtain wave functions and energy spectra. To compute the differential conductance in the tunneling spectroscopy setup of Fig. 2(a) we use the scattering formalism. The scattering matrix, relating incoming and outgoing modes in the normal lead, is

$$S = \begin{bmatrix} S_{\text{ee}} & S_{\text{eh}} \\ S_{\text{he}} & S_{\text{hh}} \end{bmatrix}, \tag{7}$$

where $S_{\alpha\beta}$ is the block of the scattering matrix with the scattering amplitudes of incident particles of type $\beta$ to outgoing particles of type $\alpha$. The differential conductance is

$$G(E) = \frac{dI}{dV} = \frac{e^2}{h}(N_{\text{e}} + T_{\text{he}} - T_{\text{ee}}), \tag{8}$$

with $N_{\text{e}}$ the number of propagating electron modes in the lead and $T$ the transmissions that are related to the scattering matrix by

$$T_{\alpha\beta}(E) = \text{Tr}\left\{\left[S_{\alpha\beta}(E)\right]^{\dagger} S_{\alpha\beta}(E)\right\}. \tag{9}$$

We obtain the discretized Hamiltonian and the scattering matrix (7) numerically using Kwant [30], see the supplementary material for source code and data [31]. We use adaptive parallel sampling of functions by using the Adaptive package [32].

### 2.3 Couplings from Mahaux-Weidenmüller formula

We investigate how the low-energy states in the proximitized nanowire couple to the propagating electron modes in the normal lead in the setup of Fig. 2(a), since this coupling determines the conductance through the lead-wire interface. To do so, we write the scattering matrix Eq. (7) in a different form using a generalized form of the Mahaux-Weidenmüller formula derived in Ref. [33]:

$$S(E) = \mathbb{1} - 2\pi i W \left(E - H + i\pi W^{\dagger}W\right)^{-1} W^{\dagger}. \tag{10}$$

Here, $H$ is the modified Hamiltonian of the scattering region, $E$ is the excitation energy, and $W$ is the matrix containing couplings of the lead modes to the states in the scattering region.

To compute the coupling to the lowest energy eigenstate of the Hamiltonian, $\psi_+(E)$, and its particle-hole symmetric partner $\psi_-(-E) = \mathcal{P}\psi_+(E)$, with $\mathcal{P}$ the particle-hole operator, we introduce a matrix $P = [\psi_+, \psi_-]$. The product $WP$ contains the coupling of the lead modes to the pair of lowest-energy eigenstates of the Hamiltonian $H$. To calculate the coupling to the Majorana components $\psi_1, \psi_2$, we write $\psi_1, \psi_2$ as linear combinations of $\psi_+, \psi_-$,

$$\begin{bmatrix} \psi_1 \\ \psi_2 \end{bmatrix} = \begin{bmatrix} e^{i\phi} & e^{-i\phi} \\ ie^{i\phi} & -ie^{-i\phi} \end{bmatrix} \begin{bmatrix} \psi_+ \\ \psi_- \end{bmatrix} = U \begin{bmatrix} \psi_+ \\ \psi_- \end{bmatrix} \tag{11}$$

for some arbitrary phase $\phi$. In this Majorana basis, $\psi_1$ and $\psi_2$ satisfy $\psi_1 = \mathcal{P}\psi_1, \psi_2 = \mathcal{P}\psi_2$. The projected coupling matrix $\hat{W}$ in this basis has the form

$$\hat{W} = WPU^{\dagger} = \begin{bmatrix} t_{1\uparrow} & t_{1\downarrow} & t_{1\uparrow}^* & t_{1\downarrow}^* \\ t_{2\uparrow} & t_{2\downarrow} & t_{2\uparrow}^* & t_{2\downarrow}^* \end{bmatrix}^T, \tag{12}$$

where $t_{\gamma\sigma}$ is the coupling of Majorana component $\gamma = 1, 2$ to a lead electron mode with spin $\sigma = \uparrow, \downarrow$, the complex conjugate $t^*_{\gamma\sigma}$ the coupling to the corresponding lead hole modes, and with $t^2$ in units of energy. We choose the phase $\phi$ such that it minimizes the off-diagonal elements $t_{1,\downarrow}, t_{2,\uparrow}$, which results in Majorana components with opposite spin. The computation of the coupling matrix $W$ from the propagating modes in the lead as computed with Kwant [30] is done using the method of Ref. [33].

## 2.4 Analytic conductance expressions in different coupling limits

The anti-alignment of the Majorana spins allows for an analytic expression of the conductance Eq. (8). The Hamiltonian in the Majorana basis $\{\psi_1, \psi_2\}$ reads

$$H_{\mathrm{M}} = \begin{bmatrix} 0 & iE_{\mathrm{M}} \\ -iE_{\mathrm{M}} & 0 \end{bmatrix}, \tag{13}$$

with $E_{\mathrm{M}}$ the coupling energy between $\psi_1$ and $\psi_2$. When the spins of the Majorana components are anti-parallel, the projected coupling matrix Eq. (12) simplifies to

$$\hat{W} = \begin{bmatrix} t_1 & 0 & t^*_1 & 0 \\ 0 & t_2 & 0 & t^*_2 \end{bmatrix}^T, \tag{14}$$

where $t_1 \equiv t_{1,\uparrow}$ and $t_2 \equiv t_{2,\downarrow}$. For subgap energies, only Andreev reflection processes contribute to conductance, simplifying Eq. (8) to

$$G(E) = \frac{2e^2}{h} \mathrm{Tr}\left( \left[ S^{eh} \right]^\dagger S^{eh} \right). \tag{15}$$

To evaluate this expression, we substitute Eqs. (13) and (14) into Eq. (10) and take out the electron to hole scattering block $S^{eh}$ (see Eq. (7)). To further simplify the resulting expression, we define coupling energies $\Gamma_1 = 2\pi t^2_1$ and $\Gamma_2 = 2\pi t^2_2$, [34] and study the regime $\Gamma_1 \gg E_{\mathrm{M}}, \Gamma_2$, which describes one strongly coupled and one weakly coupled low-energy state. This approximation yields

$$G(E) \approx \frac{2e^2}{h} \left( \frac{\Gamma_1^2}{\Gamma_1^2 + E^2} + \frac{\Gamma_2^2 - 2E_{\mathrm{M}}^2 \Gamma_2 / \Gamma_1}{\Gamma_2^2 + 2E_{\mathrm{M}}^2 \Gamma_2 / \Gamma_1 + E^2} \right), \tag{16}$$

see App. A for a derivation. So, Eq. (16) gives the subgap conductance through an NS interface expressed in three energy parameters $\Gamma_1, \Gamma_2$ and $E_{\mathrm{M}}$.

Equation (16) is a sum of two (semi-)Lorentzian functions, both with a peak height of $2e^2/h$. In the limit $\Gamma_1 \gg \Gamma_2, E_{\mathrm{M}}$, the first Lorentzian, with a peak width of $\sim \Gamma_1$, is much broader than the second Lorentzian of peak width $\sim \Gamma_2$. The second, narrower Lorentzian is positive for $\Gamma_2 > 2E_{\mathrm{M}}^2 / \Gamma_1$ and negative for $\Gamma_2 < 2E_{\mathrm{M}}^2 / \Gamma_1$, and hence respectively increases the conductance around $E = 0$ to $4e^2/h$ or decreases it to 0, depending on the coupling strength of the second low-energy state. This result explains the numerical findings of Ref. [18]. When $\Gamma_2 \ll E_{\mathrm{M}}^2 / \Gamma_1$, the curve shape is similar to the single-mode result of Ref. [35]. Temperature broadens the Lorentzian peaks, therefore the second peak is experimentally only observable when $k_{\mathrm{B}} T \lesssim \Gamma_2$. Therefore, in the limit $\Gamma_1 \gg \Gamma_2, E_{\mathrm{M}}$, zero-bias conductance is quantized to $2e^2/h$ provided $k_{\mathrm{B}} T > \Gamma_2$. Upon increasing $\Gamma_2$, or decreasing temperature, an additional, narrower zero-bias peak is observable, either positive and increasing the overall conductance to $4e^2/h$ or negative and decreasing it to zero, depending on the sizes of $\Gamma_1, \Gamma_2$ and $E_{\mathrm{M}}$. When both $\Gamma_{1,2} \lesssim k_{\mathrm{B}} T$, both zero-bias conductance peaks are not observable, resulting in a zero subgap conductance.

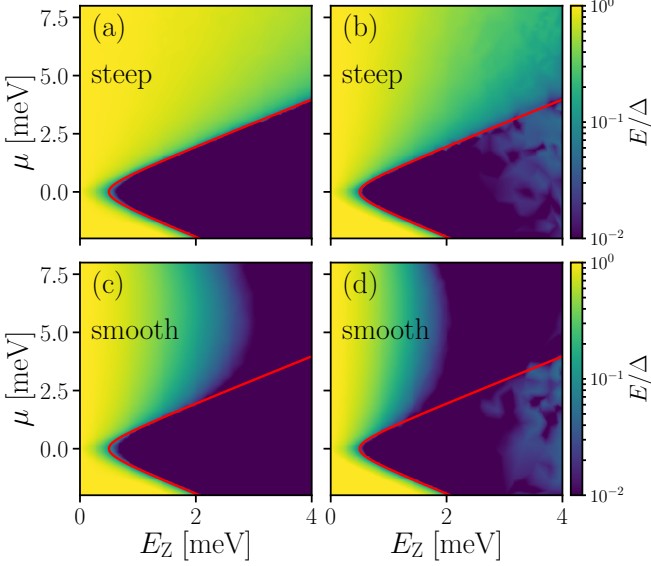

Figure 3: Phase diagram as a function of $E_Z$ and $\mu$ of the device sketched in Fig. 2(a), for (a) $\alpha = 100\,\text{meV}\,\text{nm}$, $\sigma = 10\,\text{nm}$, (b) $\alpha = 40\,\text{meV}\,\text{nm}$, $\sigma = 10\,\text{nm}$, (c) $\alpha = 100\,\text{meV}\,\text{nm}$, $\sigma = 200\,\text{nm}$, and (d) $\alpha = 40\,\text{meV}\,\text{nm}$, $\sigma = 200\,\text{nm}$. The red line indicates the topological phase boundary $E_Z = \sqrt{\Delta^2 + \mu^2}$. The color indicates the lowest energy of the Hamiltonian in units of $\Delta$ on a logarithmic scale. The potential barrier height is $V = 10\,\text{meV}$.

# 3 Phase diagram, wave functions and couplings of quasi-Majoranas

Earlier works have presented Hamiltonian spectra as a function of the magnetic field, where for specific parameter choices quasi-Majorana states occur in the trivial regime [15, 19]. To investigate more systematically in which parameter ranges these states occur, we compute a phase diagram as a function of Zeeman energy $E_Z$ and chemical potential $\mu$. To do so, we consider the system Fig. 2(a), decoupled from the lead by introducing a hard-wall boundary condition at $x = 0$ (sufficiently far into the barrier such that quasi-Majoranas can still form at the barrier slope). We compute the energy of the lowest eigenstate of Hamiltonian (3) as a function of $E_Z$ and $\mu$, see Fig. 3. In all four panels, inside the topological phase (red line), the lowest energy of the Hamiltonian is exponentially small, indicating the existence of a zero-energy state. This zero-energy state only exists in the topological phase for Fig. 3(a) and (b), when the potential barrier is steep. For a smooth potential, there is a large area of quasi-Majorana states with zero energy outside the topological phase, with a growing area as the SOI weakens, see Fig. 3(c) and (d).

We investigate in Fig. 4 the energy spectra and wave functions corresponding to different regimes of the phase diagrams of Fig. 3. The blue and red vertical lines in the energy spectra of Fig. 4(a) and (b) show the Zeeman energies at which the corresponding (quasi-)Majorana wave functions are plotted in the lower panels. For the parameters of the left column of the figure, we find a partially separated quasi-Majorana density of states in the trivial regime with a separation of the order of $\xi$ (Fig. 4(c)), and with opposite spin densities (Fig. 4(e)). The system goes into a topological phase for a further increase of Zeeman energy, with well-separated Majorana bound states at the system's endpoints (Fig. 4(g)).

We note that partially separated (i.e. separated on the order of the coherence length $\xi$

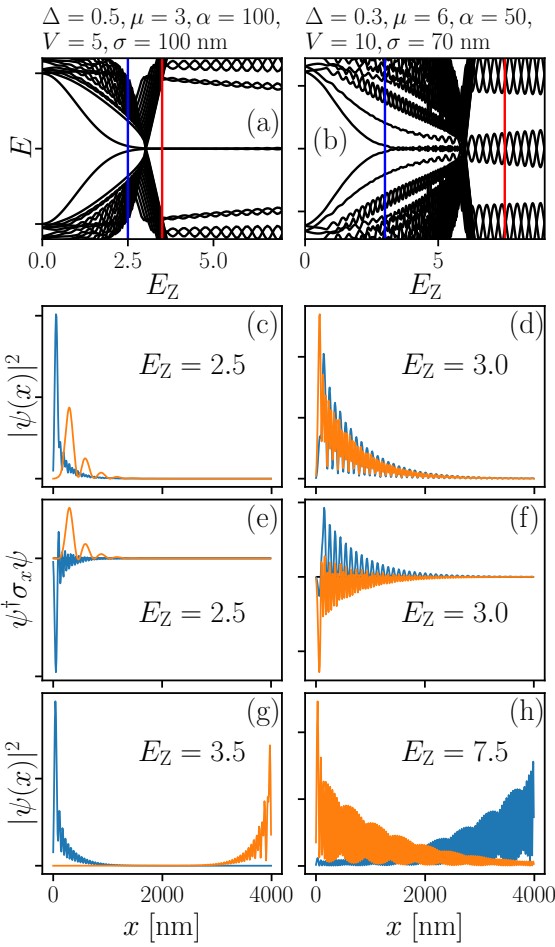

Figure 4: Wave functions and spin densities in the device sketched in Fig. 2(a) with a smooth potential on the left edge. The left column shows results for parameters such that the Majorana components in the trivial regime are partially separated, the right column for fully overlapping components. (a, b): Energy spectrum as a function of $E_Z$. The blue and the red vertical lines indicate the Zeeman energies at which respectively the trivial and topological wave functions of the lower panels are plotted. (c, d): Probability densities of the Majorana components of the lowest Hamiltonian eigenstate in the quasi-Majorana regime. (e, f): Spin densities of both Majorana components along $x$. (g, h): Majorana wave functions in the topological regime. Energy scales are given in meV, SOI strength in meV nm. The nanowire length is set to $L_{SC} = 4 \, \mu$m.

or more) have been reported in the literature before. [19, 36]. In addition to these works, we find realistic parameter regimes with quasi-Majorana states consisting of spatially *almost completely overlapping* Majorana components, with a separation much smaller than $\xi$, that still have an exponentially suppressed near-zero energy. This is shown in the right column of Fig. 4, with Fig 4(b) the corresponding energy spectrum. Figure 4(d) shows that for this choice of parameters, the quasi-Majorana components nearly completely overlap, having a displacement much smaller than $\xi$. The separation of both quasi-Majorana wave functions is governed largely by the separation of the classical turning points for the spin-split tunnel barrier, controlled by the parameters of the system (such $\mu$ or $E_Z$) and the tunnel barrier. In general, the quasi-Majoranas wave function overlap increases with decreasing SOI strength $\alpha$ and smoothness $\sigma$, and increasing barrier height $V$. As in the partially separated case, also

mostly overlapping quasi-Majorana states turn into true topological states on increasing Zeeman splitting (Fig. 4(h)).

The origin of the decoupling between two quasi-Majorana states lies in the nearly opposite expectation value of the spin in $x$-direction, $\psi^{\dagger}(x)\sigma_x\psi(x)$, which we show in Fig. 4(e) and (f). As discussed in Ref. [15], the SOI strength effectively vanishes at the turning point of the smooth potential, leading to two decoupled Majorana states with spins aligned or anti-aligned to the Zeeman field (which is oriented in the $x$-direction) at the turning point. In this work, we focus on the case of weak spin-orbit coupling, as quasi-Majoranas are more prevalent in this limit (see Figs. 3(c) and (d)). In the limit of spin-orbit length larger than the wire diameter it is justified to neglect the transverse terms of the spin-orbit coupling [37], as we do in Hamiltonian (3). Hence, we observe that the individual Majorana components have a largely uniform sign of the spin expectation value.

Because quasi-Majorana states are located on the same side of a proximitized nanowire, while topological Majorana states are separated between opposite edges, one might expect local transport measurements to distinguish between both cases. However, this is generally not the case, as shown in Fig. 1: the opposite spin of both quasi-Majorana states result in a different effective barrier, which exponentially suppresses one quasi-Majorana coupling, reproducing the coupling regime of topological Majorana states. Figure 5(a, c) show the coupling parameters for a steep potential barrier (that does not suport the formation of quasi-Majorana states), and Figure 5(b, d) for a smooth barrier with quasi-Majorana states. The energy of the lowest Hamiltonian eigenstate $E_M$ is exponentially small for increasing magnetic field only in the topological regime for a steep barrier, see Fig. 5(a), but is suppressed well before the topological phase transition for a smooth barrier with quasi-Majorana states, Fig. 5(b). Likewise, the couplings across the barrier of the Majorana components of the lowest Hamiltonian eigenstate $\Gamma_1, \Gamma_2$ are exponentially different only in the topological phase for a steep potential barrier, Fig. 5(c). However, for a smooth barrier, the couplings are approximately four orders of magnitude different already in the trivial phase, Fig. 5(d). Consistently, we find that the exponential suppression of both $E_M$ and $\Gamma_2$ is stronger in the quasi-Majorana regime than in the topological regime.

The exponential suppression of the coupling between quasi-Majoranas and the coupling of one of the quasi-Majoranas across a tunnel barrier for increasing magnetic field reproduces the topological coupling regime $\Gamma_2, E_M \ll \Gamma_1$. Hence, the conductance signatures of both the quasi-Majorana regime and the topological Majorana regime are similar. In absence of quasi-Majorana states, a zero-bias conductance peak quantized to $2e^2/h$ only develops after the topological phase transition, Fig. 5(e), while in presence of quasi-Majorana states, a quantized zero-bias peak is also present in the trivial regime, Fig. 5(f). This zero-bias peak quantized to $2e^2/h$ coincides with the exponential suppression of the coupling of one of the two zero-bias states, as expected from our analytical formula, Eq. (16). Our calculations also show a narrow conductance dip around $E = 0$ due to the coupling of the second (quasi-)Majorana $\Gamma_2$ as is consistent with Eq. (16), but this is not visible in the color scheme of Fig. 5, and experimentally not visible when $\Gamma_2 \lesssim k_B T$. Hence, a quantized $2e^2/h$ zero-bias conductance peak does not distinguish between topological Majorana states and quasi-Majorana states.

## 4  Majorana non-locality and topological Josephson effect

References [26, 27] express Majorana non-locality as the ratio between the couplings $\Gamma_1, \Gamma_2$ of the two Majorana states to a probing lead. In Ref. [26], a 'quality factor' $q = 1 - \Gamma_2/\Gamma_1$ is defined, with $q = 0$ denoting two strongly coupled local Majorana states ($\Gamma_1 = \Gamma_2$), and $q = 1$ denoting complete non-locality ($\Gamma_2 = 0$). References [26, 27] propose a coupled quantum

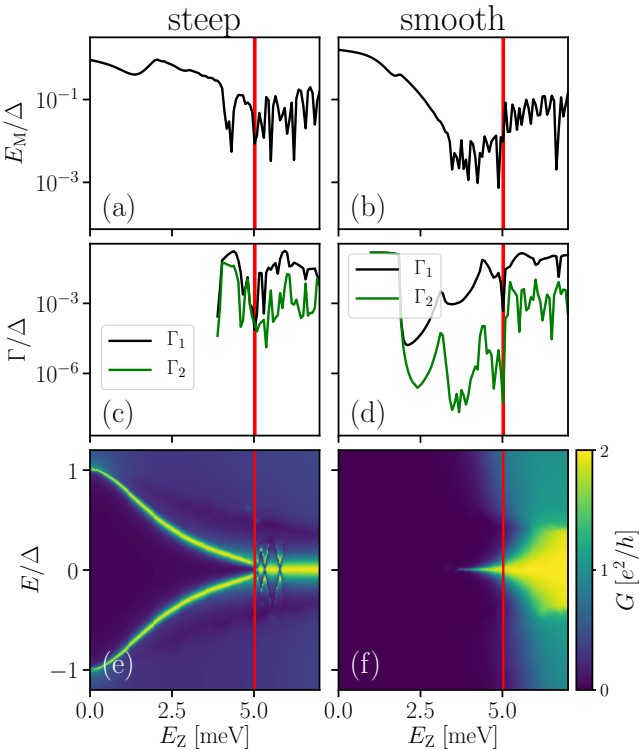

Figure 5: Coupling energy and conductance as a function of Zeeman energy $E_Z$ for a steep tunnel barrier ($\sigma = 10$ nm, left column) or for a smooth tunnel barrier ($\sigma = 100$ nm, right column) with quasi-Majorana states. (a, b): Coupling energy between the two lowest states $E_M$. (c, d): Coupling energy to the probing lead $\Gamma_1, \Gamma_2$ of the two lowest states. We do not include the coupling for small Zeeman energies, since the energy of the lowest state is too large compared to the bulk gap in this regime for the approximation of Sec. 2.3 to hold. (e, f): Conductance as a function of bias energy $E$ and Zeeman energy $E_Z$. In all panels, the red vertical line indicates the topological phase transition. The barrier heights and smoothness are $V = 18$ meV, $\sigma = 10$ nm for the left column and $V = 11.7$ meV, $\sigma = 100$ nm for the right column respectively, and the chemical potential is $\mu = 5$ meV for all panels.

dot-nanowire device, see Fig. 2(b), to determine the quality factor with a local probe, which has been experimentally implemented in Ref. [28]. The spectrum of the hybrid quantum dot-nanowire device shows anti-crossing quantum dot states and a flat zero-energy state as a function of the quantum dot chemical potential in case of well-separated Majorana states, with $E_M, \Gamma_2 \ll \Gamma_1$. When the Majorana states are closer together, the increasing coupling of the second Majorana to the quantum dot results in increasingly asymmetric diamond-like shapes in the lowest energy level across the resonance with the quantum dot states. Hence, the measurement of the energy levels in the hybrid device allows to determine the Majorana non-locality with a local probe.

Reference [22] pointed out that partially separated Andreev bound states can have different couplings to a quantum dot, mimicking the signatures of spatially separated topological Majorana states. We show that quasi-Majorana states systematically have exponentially different couplings to a quantum dot, hence the quasi-Majorana regime generally exhibits a high degree of non-locality. Figure 6(a, b) show the spectrum in the topological phase as a function of quantum dot chemical potential $\mu_{dot}$ and Zeeman energy $E_Z$ respectively. The quantum dot and the nanowire are separated by a steep barrier, so no quasi-Majorana states appear in

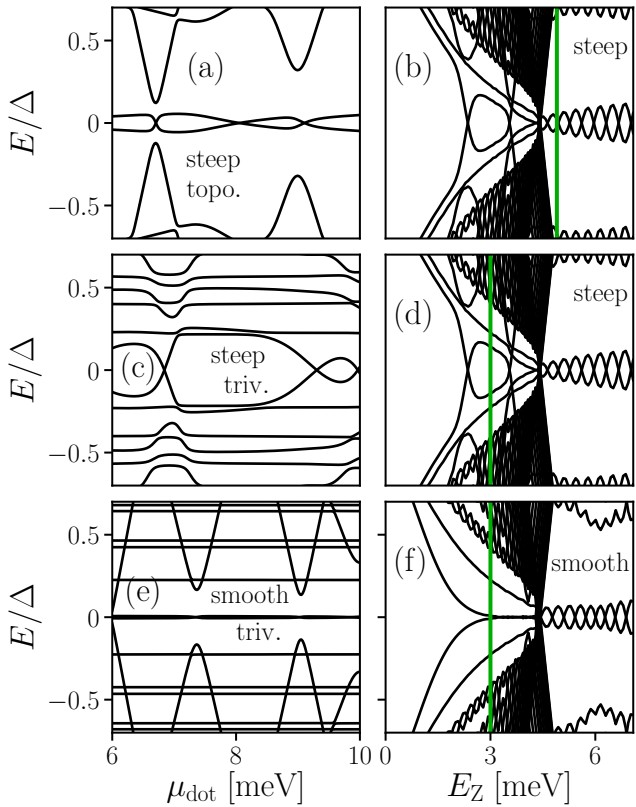

Figure 6: Energy levels in a hybrid quantum dot-nanowire device as a function of the chemical potential of the quantum dot $\mu_{dot}$ (left column), and as a function of Zeeman energy $E_Z$ (right column). Panels (a, b) show the energy levels in the topological phase with a steep barrier ($\sigma = 10$ nm), panels (c, d) in the trivial phase with a steep barrier, and panels (e, f) in the trivial phase with a smooth barrier ($\sigma = 100$ nm) in presence of quasi-Majorana states. The vertical green lines in the right panels indicate the Zeeman energy at which the corresponding left panel is computed. The barrier height and the chemical potential in all panels is $V = 9.5$ meV and $\mu = 4.4$ meV respectively.

the spectrum of Fig. 6(b). The non-locality of the Majorana states is expressed in Fig. 6(a) by the flat energy level around $E = 0$ of the non-local Majorana state, and spin-dependent anti-crossings of the quantum dot levels coupled to the local Majorana state. A flat energy level around $E = 0$ and strong anti-crossings are absent in Fig. 6(c), where the system is topologically trivial and no single Majorana state couples to the quantum dot. However, in the presence of quasi-Majoranas, Fig. 6(e, f), these characteristics occur in the trivial phase because of the exponentially different coupling of both quasi-Majorana states to the quantum dot. Therefore, since quasi-Majorana states reproduce the topological coupling regime $E_M, \Gamma_2 \ll \Gamma_1$, we observe that quasi-Majorana states can exhibit a high degree of Majorana non-locality, and consequently give rise to high quality factors, while being highly local in space. This makes Majorana non-locality and the Majorana quality factor as proposed in Refs. [26,27] unsuitable for distinguishing quasi-Majorana states from topological Majorana states.

Turning to the $4\pi$-periodic Josephson effect in a device as sketched in Fig. 2(c), we again compare a topological junction to a trivial junction with and without quasi-Majorana states. Figure 7(a) shows a $4\pi$-periodicity of the energy levels corresponding to the Majorana states located at the normal barrier, and a flat zero-energy level corresponding to the Majorana states

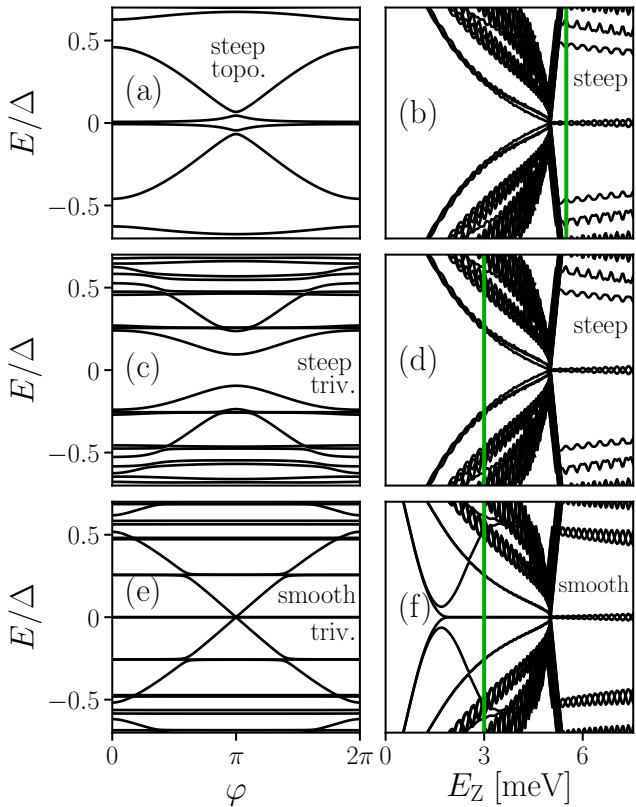

Figure 7: Energy levels in a topological Josephson junction as a function of the phase difference across the junction $\varphi$ (left column), and as a function of Zeeman energy $E_Z$ (right column). Panels (a, b) show the energy levels in the topological phase with a steep barrier ($\sigma = 10$ nm), panels (c, d) in the trivial phase with a steep barrier, and panels (e, f) in the trivial phase with a smooth barrier ($\sigma = 100$ nm) in presence of quasi-Majorana states. The vertical green lines in the right panels indicate the Zeeman energy at which the corresponding left panel is computed. The barrier height and the chemical potential in all panels is $V = 7.4$ meV and $\mu = 5$ meV respectively.

at the outer edges of the device (with a small splitting due to finite size effects), as is expected theoretically in the topological phase [24,25]. In the trivial phase, as shown in Fig. 7(c), no zero-energy state is present, and energy levels show a $2\pi$-periodicity. When the barrier is smooth, quasi-Majorana states appear in the trivial regime (see Fig. 7(f)), reproducing the flat zero-energy levels and $4\pi$-periodic levels that characterize the topological Josephson junction (Fig. 7(e)). Quasi-Majorana states reproduce the topological phase winding characteristics because two quasi-Majorana states strongly couple across the barrier, resulting in a $4\pi$-periodic level, and two have an exponentially suppressed coupling, resulting in a flat zero-energy level. Therefore, the measurement of a $4\pi$-periodic Josephson current is not a distinctive signature of topological Majorana states, but can be caused by quasi-Majorana states.

## 5  Distinctive signatures of a topological phase

Previously discussed measurement setups rely on Majorana modes to determine a topological phase, which makes them inherently sensitive to non-topological local low-energy states. Hence, a better strategy to distinguish a topological from a trivial phase is the measurement of a bulk

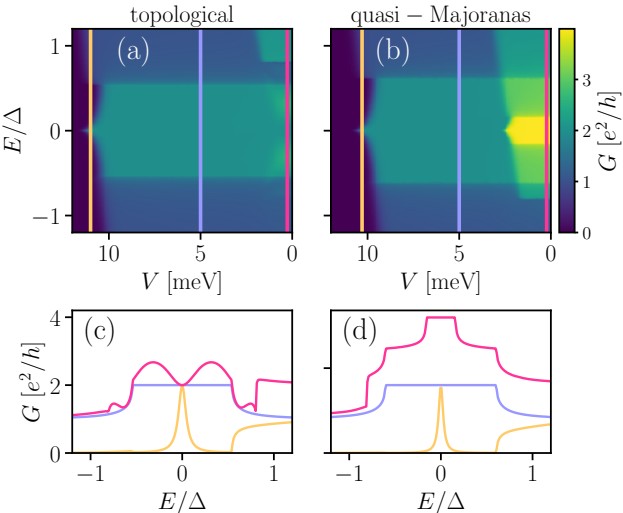

Figure 8: (a, b): Conductance as a function of bias energy $E$ and barrier height $V$ for topological Majorana states (left) and for quasi-Majorana states (right). The vertical colored lines point out the values of $V$ at which line cuts, shown in (c, d), are made. The parameter values for this plot are: $V = 7$ meV, $\mu = 4$ meV, and $\sigma = 200$ nm. Panels (a) and (c) are made at $E_Z = 4.4$ meV, in the topological phase, and panels (b) and (d) are made at $E_Z = 3.6$ meV, in the trivial phase.

phase transition rather than the measurement of individual Majorana states, which has been proposed in several earlier works. Reference [38] discusses quantized thermal conductance and electrical shot noise in a proximitized nanowire coupled to two normal leads as signatures of a topological phase transition. Reference [39] proposes the measurement of differences in conductance at one lead connected to a proximitized nanowire when changing the coupling to another lead, while Ref. [40] predicts a sign change of the spin component of bulk bands along the magnetic field as a measure of a topological phase transition. Finally, Ref. [41] proposes the detection of rectifying the behavior of the nonlocal conductance $G$ between two spatially separated leads as a function of the bias $E$, $G(E) \propto E$, as a signature of a bulk phase transition. These proposals all rely on bulk properties and therefore more reliably detect a topological phase than probing a local Majorana state, which might be mimicked by other localized low-energy states.

We also suggest an alternative approach relying on local conductance measurements that allows to distinguish topological Majorana states from quasi-Majorana states. According to Eq. (16), when the coupling of the second low-energy subgap state $\Gamma_2$ exceeds $k_B T$, an experimentally observable zero-bias conductance peak of $4e^2/h$ develops. Hence, our approach does not focus on a quantized conductance peak in the tunneling spectroscopy [42] when $\Gamma_2$ is strongly suppressed, but on a conductance measurement in the open regime. We demonstrate the effect of opening the tunnel barrier $V \to 0$ (with $V$ the height of the potential barrier given in Eq. (4)) on the conductance with true Majorana states and with quasi-Majorana states in Fig. 8, using the microscopic model (3). Figure 8(a) shows the conductance as a function of bias energy $E$ and barrier height $V$ in the topological phase with spatially separated Majorana states. In the tunneling regime, the conductance shows a zero-bias peak quantized to $2e^2/h$ (see also the light-brown line cut in panel (c)), which broadens to a plateau of $2e^2/h$ height upon opening the barrier (purple line cut). When the barrier height is further reduced, the conductance at finite bias increases due to Andreev enhancement, but stays fixed to $2e^2/h$ at zero bias due to the presence of a single Majorana state (pink line cut) [11,43]. Quasi-Majorana

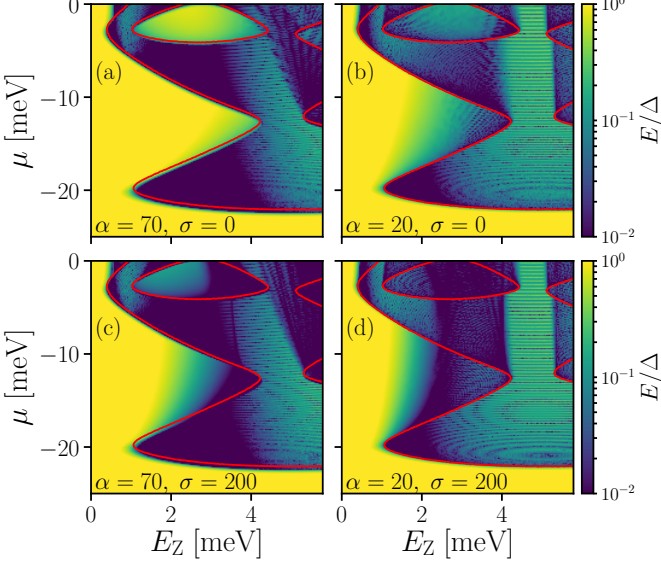

Figure 9: Phase diagram as a function of $E_Z$ and $\mu$ of the 3D device, with orbital magnetic field and an external superconductor included. The red line indicates the topological phase boundary. The color indicates the lowest energy of the Hamiltonian in units of $\Delta$ on a logarithmic scale. Different panels have different parameter values for the SOI strength $\alpha$ (in meV nm) and potential smoothness $\sigma$ (in nm). The potential barrier height is $V = 25$ meV.

states also exhibit a conductance peak of $2e^2/h$ in the tunneling regime and a conductance plateau of $2e^2/h$ in the quasi-open regime as shown in Fig. 8(b) and the line cuts in Fig. 8(d). However, upon further opening the barrier, both quasi-Majorana states couple to the lead, resulting in a conductance peak of $4e^2/h$ which broadens to a plateau when further reducing $V$.

Note that in this argument we rely on a strong coupling of both quasi-Majorana states to the lead, tunable by e.g. an external tunnel gate. Should the coupling be limited by intrinsic effects such as local disorder, the conductance may stay below $4e^2/h$. In the topological case, the conductance exceeds $2e^2/h$ for voltages away from zero. This finite bias conductance value is not universal, and depend on details of the system, such as potential shapes. In contrast, the zero-bias conductance must *always* stay quantized at $2e^2/h$ as long as not more than two conductance channels are opened in the tunnel barrier, due to particle-hole symmetry. [11] Therefore, while a zero-bias conductance peak or conductance plateau quantized to $2e^2/h$ does not distinguish quasi-Majorana states from topological Majorana states, a quantized zero-bias *dip* in the conductance in the open regime does.

## 6 Quasi-Majorana states in a 3D nanowire

Because quasi-Majorana states so far have been studied in one-dimensional systems [15, 16, 18, 19, 21, 22], it is uncertain how likely quasi-Majoranas are to appear in realistic situations. While currently doing a fully realistic simulation of a three-dimensional device is beyond state of the art, we do a 3D simulation that includes the orbital effect of magnetic field [44, 45], transverse spin-orbit coupling, multiple modes mixed by an inhomogeneous potential in the direction perpendicular to the wire axis, and an external superconducting shell proximitizing the nanowire (see App. B for a detailed description of the model).

We show the phase diagram of this 3D device as a function of $\mu$ and $E_Z$ in Fig. 9. The upper panels, with a steep potential barrier ($\sigma \rightarrow 0$), show that the emergence of a zero-energy state coincides with the topological phase, which has a more complicated shape compared to Fig. 3 due to multiple modes and the orbital effect of the magnetic field. In Fig. 9(b), the gap outside the topological phase is weaker due to a weaker spin-orbit coupling, but no robust trivial zero-energy state emerges. However, Fig. 9(c, d) show that for a smooth potential barrier ($\sigma = 200$ nm) a region of zero-energy quasi-Majorana states emerges, especially prominent for weak spin-orbit strength $\alpha = 20$ meV nm, Fig. 9(d). Figure 9 is qualitatively similar to Fig. 3: for a smooth potential, regions of zero-energy quasi-Majoranas emerge, increasing in size for decreasing spin-orbit strength. Thus, we find that quasi-Majorana states are also present in realistic 3D systems with smooth potentials that are close to currently available experimental devices.

## 7 Braiding operations with quasi-Majorana states

Braiding schemes that demonstrate and utilize the non-Abelian statistics of Majorana states are subdivided into gate-controlled braiding in T-junctions [46–48], Coulomb-assisted braiding in Josephson junctions [49,50], or measurement-based braiding in topological nanowires coupled to quantum dots [51–53]. Having quasi-Majorana states in the topologically trivial phase in these devices still admits braiding. Gate-controlled braiding requires microscopically precise manipulation of electrostatic potentials, and therefore we leave gate-controlled braiding with quasi-Majorana states as a topic for future research. On the other hand, the other two schemes only rely on the coupling to individual Majorana states, which is possible in the quasi-Majorana regime, since quasi-Majorana states couple exponentially different across a tunnel barrier. The presence of the second, uncoupled quasi-Majorana state still allows these braiding schemes to work [54]. We show a possible setup for a measurement-based braiding scheme with quasi-Majorana states in Fig. 10(a), where only one quasi-Majorana state of each pair couples to the adjacent quantum dot, and for a Coulomb-assisted braiding scheme in Fig. 10(b), where only one quasi-Majorana state of each pair couples to one other quasi-Majorana state of the two other nanowires.

To estimate whether quasi-Majorana states are realistic candidates for braiding, we compare quasi-Majorana energy and length scales to braiding requirements. Coulomb-assisted and measurement-based braiding involves a fermion parity measurement in a transmon [55], where the parity shift is expressed in a resonance frequency shift $\Delta\omega$, which has been estimated in Ref. [53] for realistic parameters as $\Delta\omega \sim 100$ MHz. Hence, the transmon sensitivity must exceed 100 MHz, which limits the quasi-Majorana energy splitting to $\hbar\Delta\omega \sim 0.1\ \mu$eV. The energy splitting $E_M$ for the parameters of Fig. 5 does not meet this requirement (see Fig. 5(b)), but we find that for increasing barrier smoothness $\sigma$ and SOI strength $\alpha$ (while keeping the wire length fixed to $L_{SC} = 3\ \mu$m), the splitting is reduced to a value below the braiding requirement. As an example, for experimentally realistic values of $\sigma = 150$ nm and $\alpha = 100$ meV nm, we find a quasi-Majorana splitting of 0.1 $\mu$eV. Additionally, we consistently observe that the coupling energy $E_M$ in the quasi-Majorana regime is an order of magnitude smaller than in the topological regime. The smaller quasi-Majorana coupling compared to the topological Majorana coupling is due to the lower magnetic fields in the quasi-Majorana regime, which results in a smaller coupling of quasi-Majorana states to the other end of the wire. The suppression of the coupling of the second quasi-Majorana state to the outside is orders of magnitude smaller, $\Gamma_2 \sim 10^{-3}\ \mu$eV, and again we find this suppression stronger in the quasi-Majorana regime than in the topological regime, see Fig. 5(d). Consequently, using quasi-Majorana states may be an attractive approach to demonstrate braiding properties.

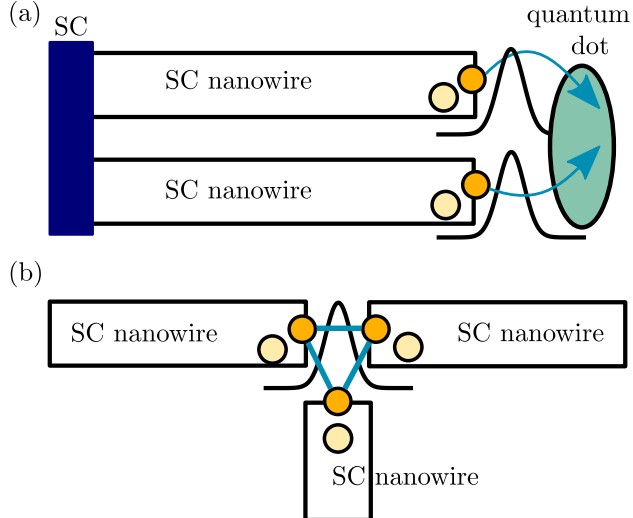

Figure 10: (a): Braiding in a measurement-based device of parallel superconducting nanowires coupled to a quantum dot. Only one of each pair of quasi-Majorana states effectively couples across the smooth barrier to the quantum dot (blue arrows). (b): Tri-junction setup for Coulomb-assisted braiding with quasi-Majorana states. From the pairs of quasi-Majoranas in each nanowire, only one couples across the tunnel barrier to the other two quasi-Majoranas (blue lines).

## 8 Summary and Outlook

Experimental setups to measure Majorana states in hybrid semiconductor-superconductor nanowire devices contain electrostatic gates that can generate smooth potential profiles, which give rise to non-topological quasi-Majorana states, that have an exponentially suppressed energy as a function of the magnetic field. Additionally, one of the quasi-Majorana states has an exponentially suppressed coupling across the tunnel barrier. This makes quasi-Majoranas mimic all local Majorana signatures, specifically a quantized zero-bias peak conductance in the tunneling spectroscopy, the resonance spectrum in a coupled nanowire - quantum dot device, and $4\pi$-periodicity of the energy levels as a function of phase in a Josephson junction. Therefore, it is impossible to categorise signatures in current Majorana experiments into topological Majorana states or trivial quasi-Majorana states.

A measurement of a bulk phase transition, rather than a measurement of the presence of local (quasi-)Majorana states, can experimentally distinguish non-topological quasi-Majorana states from topological Majorana states. Additionally, we propose to measure conductance in the open regime, which results in a plateau at $G = 4e^2/h$ around zero bias in the conductance in presence of quasi-Majoranas, and in a conductance dip to $G = 2e^2/h$ at zero bias in presence of topological, spatially separated Majoranas.

While quasi-Majorana states make it harder to unambiguously demonstrate topological Majorana states, they reproduce topological properties such as braiding. Quasi-Majorana states lack true topological protection and are hence sensitive to magnetic impurities or other short-range disorder mechanisms that break the smoothness of the potential barrier. However, due to the progress in device design, the current experimental devices are likely to be in the ballistic regime required to support robust quasi-Majorana states [56–59]. Also, for a given chemical potential, quasi-Majorana states emerge for smaller magnetic fields, which reduces the coupling to the opposite end of the wire compared to topological Majorana states, resulting in smaller energy splittings. Furthermore, combined with topological Majorana states, quasi-Majorana states increase the overall phase space in which protected quantum computing can

be performed. Therefore, it may be an interesting direction of further research to engineering quasi-Majorana states to study topological properties.

## Acknowledgments

We thank Dmitri Pikulin, Bernard van Heck, Torsten Karzig, İnanç Adagideli, Karsten Flensberg, John Watson and Leo Kouwenhoven for valuable discussions. This work was supported by ERC Starting Grant 638760, the Netherlands Organisation for Scientific Research (NWO/OCW) as part of the Frontiers of Nanoscience program, and Microsoft Corporation Station Q.

**Author contributions**   M.W. initiated the project. A.V. performed the calculations and simulations, except for the 3D simulations which were performed by B.N., with input from A.A. and M.W. A.V. wrote the paper, with input from all other authors.

## A   Analytic approximation for the NS interface conductance

To arrive at the analytic expression for subgap conductance through an NS interface with two low-energy subgap states in the coupling regime $\Gamma_1 \gg \Gamma_2, E_M$, Eq. (16), we start from the Mahaux-Weidenmüller formula for the scattering matrix $S(E)$:

$$S(E) = \mathbb{1} - 2\pi i W \left(E - H_M + i\pi W^\dagger W\right)^{-1} W^\dagger. \tag{17}$$

As stated in Eqs. (13) and (14), the low-energy Hamiltonian $H_M$ and coupling matrix $W$ of the two lowest-energy states to the normal lead have the form

$$H_M = \begin{bmatrix} 0 & iE_M \\ -iE_M & 0 \end{bmatrix}, W = \begin{bmatrix} t_1 & 0 & t_1^* & 0 \\ 0 & t_2 & 0 & t_2^* \end{bmatrix}, \tag{18}$$

with $W$ in the basis $\{\psi_{e,\uparrow}, \psi_{e,\downarrow}, \psi_{h,\uparrow}, \psi_{h,\downarrow}\}$ of propagating electron and hole modes of both spins in the normal lead, and $H_M$ in the Majorana basis $\{\psi_1, \psi_2\}$, where $\psi_1$ and $\psi_2$ have opposite spin. Substitution of Eq. (18) into Eq. (17) gives an expression for the scattering matrix $S(E)$ in terms of $E_M$ and the coupling energies $\Gamma_1, \Gamma_2$:

$$S(E) = \begin{bmatrix} S^{ee} & S^{eh} \\ S^{he} & S^{hh} \end{bmatrix} = \begin{bmatrix} 1+A & A \\ A & 1+A \end{bmatrix}, \tag{19}$$

where

$$A = \frac{1}{Z} \begin{bmatrix} i\Gamma_1(E + i\Gamma_2) & -E_M\sqrt{\Gamma_1\Gamma_2} \\ E_M\sqrt{\Gamma_1\Gamma_2} & i\Gamma_2(E + i\Gamma_1) \end{bmatrix}, \tag{20}$$

with

$$Z = E_M^2 - (E + i\Gamma_1)(E + i\Gamma_2), \tag{21}$$

and $\Gamma_i = 2\pi t_i^2$ (see also Ref. [34]). Andreev reflection of an incoming electron into an outgoing hole is described by the block of the scattering matrix $S^{eh} = A$. At subgap energies, the Andreev conductance is given by

$$G(E) = \frac{2e^2}{h} \mathrm{Tr}\left(\left[S^{eh}\right]^\dagger S^{eh}\right). \tag{22}$$

In the limit $\Gamma_1 \gg E_M, \Gamma_2$, Eq. (21) is approximated by $Z \approx E_M^2 + \Gamma_1\Gamma_2 - E^2 - iE\Gamma_1$, and hence

$$|Z^2| = \left(E_M^2 - E^2 + \Gamma_1\Gamma_2\right)^2 + E^2\Gamma_1^2 \approx E_M^4 + E^4 + \Gamma_1^2(\Gamma_2^2 + E^2) + 2E_M^2(\Gamma_1\Gamma_2 - E^2). \tag{23}$$

We insert this in Eq. (20) and work out the trace of Eq. (22) using $S^{eh} = A$, which follows from Eq. (19). This yields

$$G(E) \approx \frac{2e^2}{h} \frac{2E_{\mathrm{M}}^2 \Gamma_1 \Gamma_2 + 2\Gamma_1^2 \Gamma_2^2 + E^2(\Gamma_1^2 + \Gamma_2^2)}{E_{\mathrm{M}}^4 + E^4 + \Gamma_1^2(\Gamma_2^2 + E^2) + 2E_{\mathrm{M}}^2(\Gamma_1 \Gamma_2 - E^2)}. \tag{24}$$

In the limit $\Gamma_1 \gg \Gamma_2, E_{\mathrm{M}}$, square terms in $\Gamma_1$ dominate, hence we neglect the other terms in the numerator of Eq. (24). This results in a conductance expression for the limit $\Gamma_1 \gg \Gamma_2, E_{\mathrm{M}}$:

$$G(E) \approx \frac{2e^2}{h} \frac{\Gamma_1^2(2\Gamma_2^2 + E^2)}{E_{\mathrm{M}}^4 + E^4 + \Gamma_1^2(\Gamma_2^2 + E^2) + 2E_{\mathrm{M}}^2(\Gamma_1 \Gamma_2 - E^2)}. \tag{25}$$

Next, we consider the high- and low-energy regimes separately. In the high-energy limit, $\Gamma_1, E \gg E_{\mathrm{M}}, \Gamma_2$, Eq. (25) reduces to

$$G(E) \approx \frac{2e^2}{h} \frac{\Gamma_1^2}{\Gamma_1^2 + E^2}. \tag{26}$$

Turning to the low-energy limit, $\Gamma_1 \gg E_{\mathrm{M}}, \Gamma_2, E$, we further simplify Eq. (23) to

$$Z^2| \approx \Gamma_1 \Gamma_2 \left( \Gamma_1 \Gamma_2 + 2E_{\mathrm{M}}^2 \right) + E^2 \Gamma_1^2. \tag{27}$$

The correction around zero energy to Eq. (25) is given by

$$G(E) - \frac{2e^2}{h} \approx \frac{2e^2}{h} \frac{\Gamma_2^2 - 2E_{\mathrm{M}}^2 \Gamma_2/\Gamma_1}{\Gamma_2^2 + E^2 + 2\Gamma_2 E_{\mathrm{M}}^2/\Gamma_1}. \tag{28}$$

Summing Eqs. (26) and (28) gives a simplified expression for the conductance in the limit $\Gamma_1 \gg \Gamma_2, E_{\mathrm{M}}$ at all energies, expressed in two (semi-)Lorentzian functions:

$$G(E) \approx \frac{2e^2}{h} \left( \frac{\Gamma_1^2}{\Gamma_1^2 + E^2} + \frac{\Gamma_2^2 - 2E_{\mathrm{M}}^2 \Gamma_2/\Gamma_1}{\Gamma_2^2 + 2E_{\mathrm{M}}^2 \Gamma_2/\Gamma_1 + E^2} \right). \tag{29}$$

This describes a Lorentzian of height $2e^2/h$ and width $\sim \Gamma_1$, with an additional Lorentzian with the same height $2e^2/h$ and a much narrower width $\sim \Gamma_2$ (since $\Gamma_1 \gg \Gamma_2$). This second, narrower Lorentzian is positive when $\Gamma_2 > 2E_{\mathrm{M}}^2/\Gamma_1$ and negative for $\Gamma_2 < 2E_{\mathrm{M}}^2/\Gamma_1$.

## B  Three-dimensional nanowire model

In order to verify that our conclusions still hold in three dimensions, we apply the effective low-energy model [25,29] of a semiconducting nanowire with spin-orbit coupling and a parallel magnetic field, covered by a superconductor, to a 3D system. We define $x$ as the direction along the wire, $y$ perpendicular to the wire in the plane of the substrate, and $z$ perpendicular to both wire and substrate. The corresponding Hamiltonian reads

$$\begin{aligned} H_{\mathrm{BdG}} &= \left( \frac{\boldsymbol{p}^2}{2m^*} - \mu + V(x, z) \right) \tau_z + \alpha \left( p_y \sigma_x - p_x \sigma_y \right) \tau_z \\ &\quad + \frac{1}{2} g \mu_{\mathrm{B}} \boldsymbol{B} \cdot \boldsymbol{\sigma} + \Delta \tau_x. \end{aligned} \tag{30}$$

Here $\boldsymbol{p} = -i\hbar\nabla + e\boldsymbol{A}\tau_z$ is the canonical momentum, where $e$ is the electron charge, and $\boldsymbol{A} = \begin{bmatrix} B_y z - B_z y, & 0, & B_x y \end{bmatrix}^T$ is the vector potential chosen such that it does not depend on $x$,

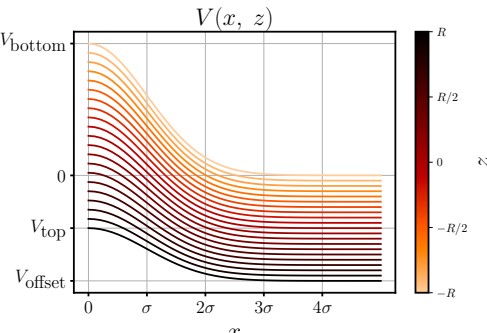

Figure 11: Potential shape inside the nanowire as given by Eq. (31). The parameter values for the simulations in Fig. 9 are $V_{\text{top}} = -30$ meV, $V_{\text{bottom}} = 25$ meV, $V_{\text{offset}} = -50$ meV, and $R = 35$ nm.

which we include in the tight-binding system using the Peierls substitution [60]. Further, $m^*$ is the effective mass, $\mu$ is the chemical potential controlling the number of occupied subbands in the wire, $\alpha$ is the strength of the SOI, $g$ is the Landé $g$-factor, $\mu_{\text{B}}$ is the Bohr magneton, and $\Delta$ is the superconducting pairing potential. The Pauli matrices $\boldsymbol{\sigma}$ and $\boldsymbol{\tau}$ act in spin space and electron-hole space respectively. We assume a Gaussian potential $V(x, z)$ inside the wire centered around $x = 0$, with different peak heights at the top ($z = R$) and bottom ($z = -R$) of the wire, and linearly interpolated for $-R < z < R$:

$$
\begin{aligned}
V(x, z) = &\, V_{\text{bottom}} \exp\left(\frac{1}{2}\frac{x^2}{\sigma^2}\right)\left(\frac{R-z}{2R}\right) \\
&+ V_{\text{offset}}\left(\frac{z+R}{2R}\right) \\
&+ \left(V_{\text{top}} - V_{\text{offset}}\right)\exp\left(\frac{1}{2}\frac{x^2}{\sigma^2}\right)\left(\frac{z+R}{2R}\right),
\end{aligned}
\tag{31}
$$

where $R$ is the wire radius, $V_{\text{bottom}}$ and $V_{\text{top}}$ are the heights of the Gaussian peaks at the bottom and top respectively, $V_{\text{offset}}$ is the difference in potential between the top and bottom, and $\sigma$ the width of the peaks. We perform numerical simulations of the Hamiltonian (30) on a 3D lattice using Kwant [30]. The source code and the specific parameter values are available in the Supplemental Material [31]. The full set of materials, including the computed raw data and experimental data, is available in Ref. [31].

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
