# Peer review of "Reproducing topological properties with quasi-Majorana states"

_SciPost Physics, doi:SciPost Phys. 7, 061 (2019)_

## Round 2 · Referee Report · Anonymous · 2019-5-4

Strengths

1- discusses braiding operations with quasi-Majorana states
2- proposes of a new distinctive signature of a topological phase (based on conductance measurements in the open regime)
3- derives of an explicit expression for the low-energy conductance

Weaknesses

1- the main idea is not new
2- some claims (e.g., regarding the smoothness of the effective potential, the spin of the quasi-Majorana states, the spatial properties on the quasi-Majorana wave functions, the conductance in the open regime) may require amendment/clarification.

Report

This manuscript studies the properties of low-energy states emerging in Majorana superconductor-semiconductor devices in the topologically-trivial regime due to the presence of a soft effective potential. The work is part of a series of studies (starting with Ref. 15 and including several recent works) dedicated to non-topological, quasi-Majorana states that mimic the (local) phenomenology of topologically-protected Majorana zero modes. The main idea (i.e. that quasi-Majorana states generated in the presence of a smooth potential can reproduce all the local properties of topological Majorana states) was discussed extensively in other works. In my opinion, the new contributions of this study include the discussion of braiding operations with quasi-Majorana states, the proposal of a new distinctive signature of a topological phase (based on conductance measurements in the open regime), and the derivation of an explicit expression for the low-energy conductance. Given the potential experimental relevance of the quasi-Majorana states, I believe that the manuscript contains enough new elements to warrant its publication. There are, however, a few points that need clarification.

1-Is “smoothness” a strong requirement for the effective potential? For example one can imagine a potential with a “sharp” step-like variation in the vicinity of the chemical potential. Would it be inconsistent with the presence of quasi-Majorana states? In this context, it would be helpful to state explicitly that all quasi-Majorana states discussed in this work are generated by a smooth potential (i.e., including the case of a proximitized wire coupled to a quantum dot; here, the dot plays no role in the emergence of the quasi-Majorana state).

2-The authors emphasize the importance of the two quasi-Majoranas having (nearly) opposite spins. How is the “spin” of the Majorana mode defined? Which of the statements regarding the spin of the quasi-Majorana states will hold (or not hold) in the presence of strong transverse spin-orbit coupling?

3-It is pointed out that the quasi-Majorana states can be either partially separated or spatially fully overlapping. However, the examples provided in Fig. 4 appear qualitatively the same. In both panel (a) and panel (b) the wave functions have a substantial overlap, with the leftmost peaks being slightly displaced by something on the order of 1/k_F. The difference between the two situations is quantitative, k_F being much larger in (b). Without additional evidence, the statement regarding the spatial properties of the quasi-Majoranas should be amended. Also, for completeness, the low-energy spectra corresponding to wave functions shown in Fig. 4 should be provided (as function of E_Z).

4-Regarding the proposed distinctive signature of the topological phase, I could imagine some potential issues. First, the authors assume that the effective potential at the end of the wire can be made perfectly flat. It is not obvious that this is the case in experiment. In other words, it may be possible that the system enters the “open regime” while a significant potential inhomogeneity persists near the end of the wire. This inhomogeneity may still support low-energy “trivial” states that produce a conductance peak lower than 4e^2/h. On the other hand, I am not sure whether or not the contributions to the differential conductance from states above the induced gap where properly accounted for. In principle, these contributions could “flood” the gap and mask the signatures discussed by the authors.

5-Finally, a few minor observations. The couplings defined below Eq. (13) do not have the appropriate dimension (some density of states is missing). The quasi-Majorana states are not topologically protected; one should be careful when stating that they open “an alternative route towards (…) topological quantum computation.” It is stated that “quasi-Majorana states emerge for smaller magnetic fields (…) resulting in smaller energy splittings.” This is true for a given chemical potential (away from the bottom of a confinement-induced sub-band), not in general.

Requested changes

Clarifications/changes that address points 1-5 of the report.

  • validity: high
  • significance: high
  • originality: good
  • clarity: high
  • formatting: excellent
  • grammar: excellent

Author:  Anton Akhmerov  on 2019-05-16  [id 515]

(in reply to Report 1 on 2019-05-04)
Category:
question

Dear referee,

We thank you for your careful analysis of our work. In the resubmitted version we would like to do our best in addressing your feedback, and for that we would like to ask for a clarification.

In your report you write:

"The main idea (i.e. that quasi-Majorana states generated in the presence of a smooth potential can reproduce all the local properties of topological Majorana states) was discussed extensively in other works."

We recognize that quasi-Majorana states are an active topic, and therefore it is important to give proper credit to the prior works. We aimed to do that in the manuscript.

To the best of our understanding the following statements are true:

  1. Before our work it was known that quasi-Majoranas may reproduce some local signatures of Majorana bound states (Refs. 15–23 of the manuscript).
  2. There were also proposals to distinguish quasi-Majoranas from Majoranas relying on a tunneling measurement that our manuscript invalidates (Refs. 26–28 of the manuscript).
  3. That the tunnel couplings of quasi-Majorana states are exponentially different in the barrier smoothness was not demonstrated.
  4. It was therefore not known that quasi-Majorana states mimic all local tunneling signatures of Majoranas.

In our view, our main result is to make the observation of the exponentially different couplings of quasi-Majoranas, and through that to conclude that quasi-Majoranas reproduce all local tunneling Majorana signatures.

If we have overlooked something, we would be happy to amend the discussion of the prior literature following specific suggestions.

Anonymous on 2019-05-24  [id 527]

(in reply to Anton Akhmerov on 2019-05-16 [id 515])
Category:
answer to question

I think that the authors have correctly identified the main aspects of the Majorana versus quasi-Majorana problem. The cited works are representative for the relevant developments in this area (and I do not believe that an exhaustive list is necessary).

---

## Round 3 · List of Changes

We thank the referee for the favorable assessment of our manuscript. We have addressed
the requests for minor clarifications raised by the referee:

1.) Indeed, the smoothness requirement is not absolute. We now refer explicitly to
previous works that have found near-zero energy states with potentials involving
kinks or abrupt changes. We have also clarified that it is indeed the smooth potential
slope of the tunnel barrier that is responsible for the appearance of quasi-Majorana
states in all of our systems, and that the quantum dot in system Fig. 2(b) serves
only as a probe.

2.) We have now clarified that with "spin" we mean the expectation value of
sigma_x.
Furthermore, note that quasi-Majorana states are actually favored by
weak spin-orbit interaction, as shown in Fig. 2(c) and (d). For this reason,
we focus on the case of weak spin-orbit interaction, where the transverse
part of the spin-orbit coupling is of negligible influence. In fact, in the
3D model of Sec. VI we do include the transverse part explicitly. For the weak
spin-orbit strengths considered there, quasi-Majorana states appear. We have
extended the discussion in this regard in the paper.

3.) We have now clarified that we compare the separation of the quasi-Majorana
states to the coherence length \xi. Furthermore, we have added to Fig. 4 the spectrum
corresponding to the wave functions as suggested by the referee.

4.) In relation to the questions raised by the referee, we would like to point out that
Fig. 8 was computed using our microscopic model, i.e. the contributions of above-gap
states are explicitly included. We have clarified this aspect in the text.
Furthermore, we have now added at the end of Section V an extended discussion on how
robust the different features of the conductance shown in Fig. 8 are.

5.) We have clarified that the coupling elements t in Eq. (12) and (14) are in units of sqrt(energy).
With respect to quasi-Majoranas allowing for topological operations, we would like to
point out that in Sec. VIII we do discuss the fact that quasi-Majoranas themselves are
not topologically protected. To avoid confusion, we have renamed topological quantum
computation with quasi-Majorana states to protected quantum computation.
With respect to the occurrence of quasi-Majoranas at smaller fields, we have now
added the clarification that this indeed holds for a given chemical potential.

You are currently on this page

Resubmission 1806.02801v3 on 14 October 2019

---

## Editorial Decision

published